# Extensive introgression and mosaic genomes of Mediterranean endemic lizards

Weizhao Yang 1✉, Nathalie Feiner 1, Catarina Pinho 2, Geoffrey M. While[3], Antigoni Kaliontzopoulou 2,
D. James Harris[2], Daniele Salvi 4 & Tobias Uller 1✉

The Mediterranean basin is a hotspot of biodiversity, fuelled by climatic oscillation and geological change over the past 20 million years. Wall lizards of the genus *Podarcis* are among the most abundant, diverse, and conspicuous Mediterranean fauna. Here, we unravel the remarkably entangled evolutionary history of wall lizards by sequencing genomes of 34 major lineages covering 26 species. We demonstrate an early (>11 MYA) separation into two clades centred on the Iberian and Balkan Peninsulas, and two clades of Mediterranean island endemics. Diversification within these clades was pronounced between 6.5–4.0 MYA, a period spanning the Messinian Salinity Crisis, during which the Mediterranean Sea nearly dried up before rapidly refilling. However, genetic exchange between lineages has been a pervasive feature throughout the entire history of wall lizards. This has resulted in a highly reticulated pattern of evolution across the group, characterised by mosaic genomes with major contributions from two or more parental taxa. These hybrid lineages gave rise to several of the extant species that are endemic to Mediterranean islands. The mosaic genomes of island endemics may have promoted their extraordinary adaptability and striking diversity in body size, shape and colouration, which have puzzled biologists for centuries.

---

[1] Department of Biology, Lund University, Lund, Sweden. [2] CIBIO/InBIO Research Centre in Biodiversity and Genetic Resources, University of Porto, Campus Agrário de Vairão, Vairão, Portugal. [3] School of Natural Sciences, University of Tasmania, Sandy Bay, Tasmania, Australia. [4] Department of Health, Life and Environmental Sciences, University of L'Aquila, Coppito, L'Aquila, Italy. ✉email: weizhao.yang@biol.lu.se; tobias.uller@biol.lu.se

Hybridization is a powerful source of genetic variation. While hybrids are usually rare in vertebrates, and may be unfit, back-crossing with parental lineages can enable transfer of adaptively relevant alleles between lineages that otherwise remain distinct[1,2]. The establishment of evolutionarily independent lineages with more evenly shared ancestry is considered exceptional, and the origin and evolutionary potential of these hybrid lineages are highly contentious[2–6]. Reticulated evolution can be an important feature of adaptive radiation[7] (e.g. *Heliconius* butterflies[8] and cichlid fish[9,10]), but it does not need to be restricted to rapidly evolving clades. Evolution of reproductive isolation is a protracted process[11], allowing genetic exchange between lineages to be a persistent feature of adaptation and diversification of a clade. This is particularly the case for organisms evolving in regions like the Mediterranean, where geologic and climatic change have caused repeated range contraction and expansion over millions of years. Still, it remains to be seen to what extent introgressive hybridization has contributed to the exceptional levels of biodiversity and endemism of the Mediterranean fauna.

In this study, we reveal the entangled evolutionary history of wall lizards, among the most abundant, conspicuous and charismatic animals of the Mediterranean. Island endemic species in particular are strikingly variable in colouration and morphology (Fig. 1), a feature that influenced early theorizing about the origins of species and adaptation[12,13]. Using whole-genome sequences of representatives of all major lineages of the *Podarcis* genus, we show that genetic exchange has been pervasive throughout the history of this clade, resulting in lineages with highly mosaic genomes that contributed to the diversity and endemism of the Mediterranean fauna.

## Results

### Genome sequencing and genetic diversity.

Genomic DNA of 34 *Podarcis* wall lizards from different lineages, including 26 recognized species, were sequenced on an Illumina platform (Supplementary Table 1). Sequence reads were aligned to the *P. muralis* reference genome[14] and variants called, generating a total of 28.4 million single-nucleotide variants (SNVs; Supplementary Fig. 1). The mean nucleotide diversity ($\pi$) for all individuals was $10.6 \times 10^{-3}$ (range from $1.7 \times 10^{-3}$ to $29.0 \times 10^{-3}$; Supplementary Fig. 2). A principal component analysis (PCA) based on genetic distance separated all individuals into four distinct geographic species clusters (Balkan group, Iberian group, Sicilian-Maltese group and Western Islands group, where the latter

includes the Balearic islands, Corsica and Sardinia), and two species that formed separate clusters (lineages of the Italian species *P. siculus* and the widely distributed species *P. muralis*; Supplementary Fig. 3).

### The evolutionary history of wall lizards.

Past attempts to construct a species tree for *Podarcis* have yielded unstable topologies, and this lack of a fully resolved phylogeny has limited our understanding of their evolutionary history[15,16]. To bridge this gap, we constructed a phylogenetic framework for wall lizards by adopting two approaches (Supplementary Table 2). We first generated two concatenated datasets by combining the SNVs from whole-genome sequence (WGS) and protein coding sequence (CDS) data, respectively, for all individuals, and inferred the phylogenies for the two datasets using a maximum likelihood (ML) approach. To alleviate biases due to the concatenation of loci with variable evolutionary histories, we complemented this strategy with a multispecies coalescence (MSC) approach. To this end, we divided the WGS data into small non-overlapping windows (200, 100, 50, 25, 10 and 5 kbs), inferred local phylogenies for each window, and reconstructed the consensus tree.

The phylogenies generated from the concatenation and MSC approaches were highly consistent (average normalized Robinson–Foulds pairwise tree distance: 5.21%; Fig. 2a, b), and clearly supported the geographic clusters of the lineages identified by the principal component analysis (Supplementary Fig. 3). The Iberian group and *P. muralis* formed a sister taxon to *P. siculus*, while the Balkan group and Sicilian-Maltese group formed a sister taxon to the Western Islands group. The topology among these major groups was supported by all phylogenies, although some species within the Iberian (e.g. *P. carbonelli*, *P. guadarramae* and *P. vaucheri*) and the Balkan (e.g. *P. peloponnesiacus* and *P. tauricus*) groups showed low support rates and conflicting topologies (Supplementary Fig. 4). Local trees based on window sizes of 200 kb produced the most stable results with the least polytomies (Supplementary Fig. 5), but also revealed extensive discordances (Fig. 2c). For example, the consensus tree topology only accounted for 8.58%, and the top 8 topologies accounted for only 31.46% of all local trees inferred from 200 kb windows (Supplementary Fig. 5). Most of the low-support nodes were found within the Iberian group. The mitochondrial DNA (mtDNA) tree was inconsistent with trees based on the nuclear genome and did not support the monophyly of the Western Islands group and the Sicilian-Maltese group (Fig. 2b, Supplementary Fig. 6, Supplementary Table 3). These discordances

*Podarcis* spp.

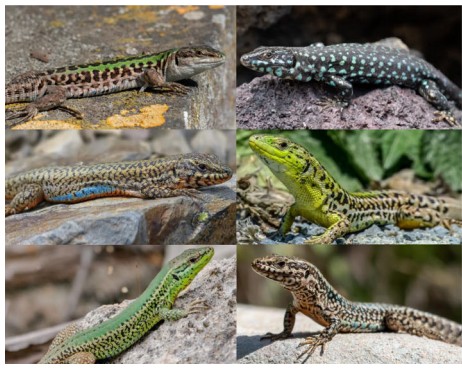

*Podarcis pityusensis*

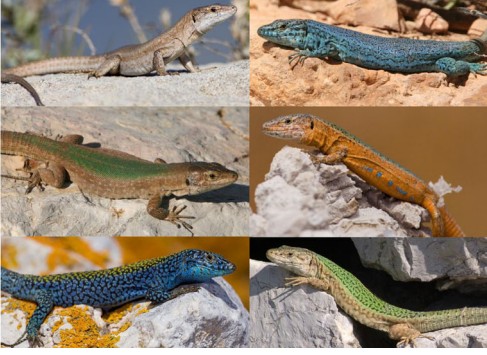

**Fig. 1 Phenotypic diversity of wall lizards.** Left panel shows representative male phenotypes of six *Podarcis* species: *P. siculus* (top left), *P. filfolensis* (top right), *P. erhardii* (middle left), *P. tauricus* (middle right), *P. waglerianus* (bottom left) and *P. muralis* (bottom right). Some *Podarcis* species are extraordinarily variable, as illustrated by the right panel that shows representative male phenotypes of six distinct populations of *Podarcis pityusensis*, a species endemic to Ibiza, Formentera and adjacent islets. Photo credits: Birgit and Peter Oefinger (https://www.eurolizards.com/) for all six species on the left panel, and for *P. pityusensis* Day's Edge Productions (top left, middle right, bottom left) and Mike Zawadzki (top right, middle left and bottom right).

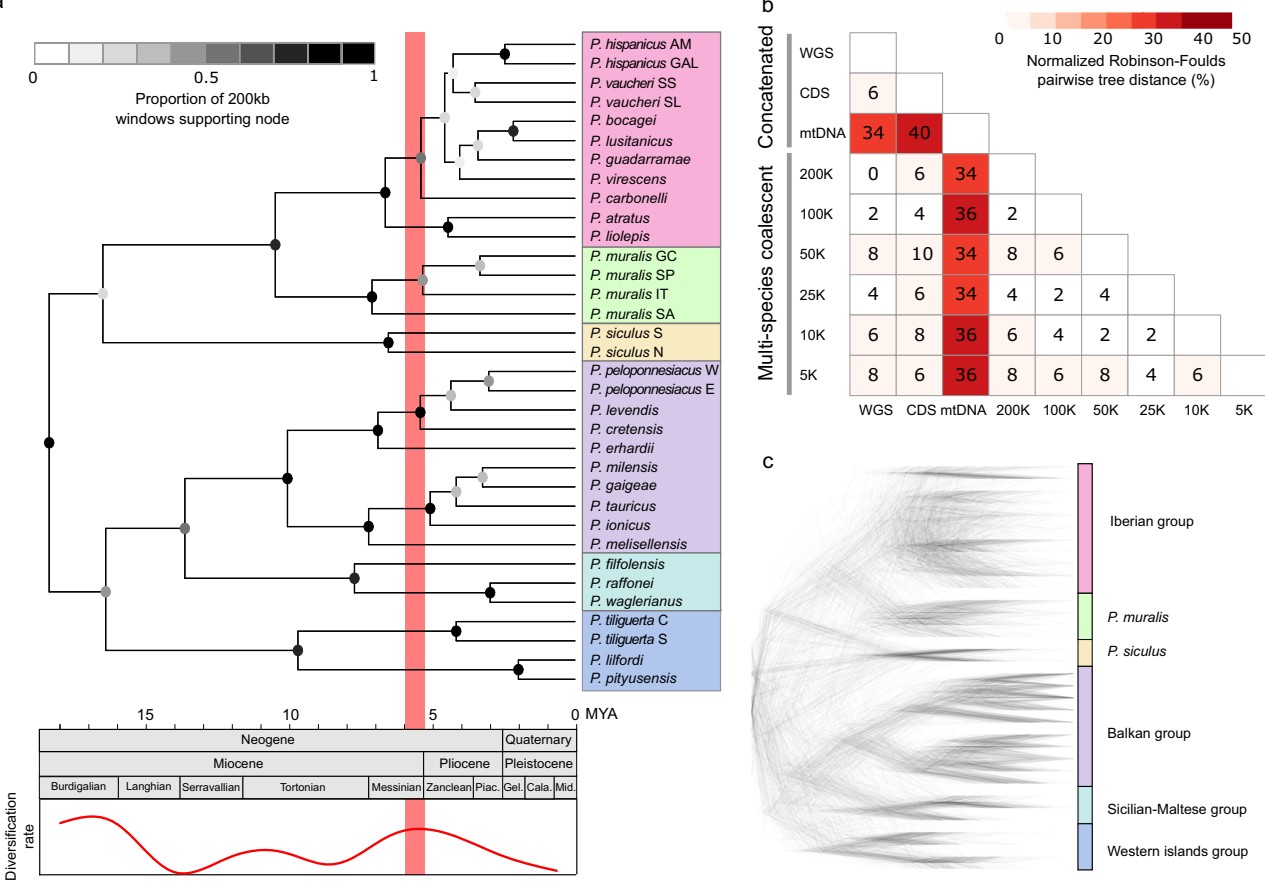

**Fig. 2 Phylogeny of *Podarcis* wall lizards. a** The time-calibrated ML phylogenetic tree of the 34 major lineages of wall lizards based on whole-genome sequences (outgroup not shown). All nodes were 100% supported with 1000 bootstrap replicates, but not across all local trees (grey-scale of nodes signifies proportion of local trees derived from 200 kb windows in support of a given node). The lower panel shows the geological time scale of the evolutionary history of *Podarcis* and the estimated diversification rate. The thick red line indicates the Messinian Salinity Crisis (~6.0–5.3 MYA). The coloured rectangles on the tips of the tree denote geographically coherent groups of wall lizards. **b** A summary of the similarities, given as normalized Robinson–Foulds distances, between all phylogenies reported in this study (phylogenies are shown in Supplementary Fig. 5 and Supplementary Figs. 8–10). See Supplementary Table 2 for a summary of approaches and datasets. **c** A 'DensiTree' illustration of the prevalent discordances among 500 randomly selected local trees derived from 200 kb windows (with branch length proportional to genetic change). Abbreviations: Cala. Calabrian, Gel. Gelasian, Mid. Middle, Piac. Piacenzian.

between nuclear and mitochondrial DNA indicate introgression of mtDNA from geographically adjacent, but sometimes distantly related, donor lineages into the *P. hispanicus* 'GAL' lineage, into the *P. siculus* lineage, into the *P. tiliguerta* lineage and into the Sicilian subclade (Supplementary Fig. 6).

We estimated the divergence times for all lineages based on the WGS dataset by using a relaxed clock approach. To reduce the influence of extensive introgression, we used only the genomic regions for which local trees were concordant with the consensus phylogeny. Two secondary calibrations were adopted from a recent phylogeny of the Lacertidae[17]; the root node (37.55 million years ago, MYA) and the crown node of *Podarcis* (18.60 MYA). We estimated that the split among the major clades of *Podarcis* wall lizards took place in the Miocene between ~16.7–9.8 MYA (Fig. 2, Supplementary Fig. 7). To characterize the pattern of diversification, we analyzed shifts in diversification rates on a lineage through time (LTT) curve using a birth–death model[18]. The results suggested two periods of rapid diversification around 17–15 MYA (coinciding with the Burdigalian–Langhian transition) and around 6.5–4.0 MYA (coinciding with the Messinian–Zanclean transition), the latter contributing to the rich diversity of the Balkan and Iberian groups (Fig. 2, Supplementary Fig. 7).

**Rampant introgression throughout the evolutionary history of *Podarcis*.** As we found extensive discordances among local trees, we systematically tested for signatures of introgression between all lineages. We first employed the Patterson's D-statistics (ABBA-BABA test)[19] for all 5984 triplets of the 34 lineages and *Archaeolacerta bedriagae* as outgroup. The majority of the triplets (77.0%) showed significant deviation from neutrality (|Zscore| > 3.3), suggesting that the discordances among local trees and the phylogenetic uncertainties were not only caused by incomplete lineage sorting (ILS), but also provide strong evidence of past admixture between lineages (Supplementary Fig. 11). We also tested the sharing of particularly long genomic blocks between distantly related species by reconstructing co-ancestry matrices in fineSTRUCTURE[20], which revealed patterns that were consistent with the D-statistics (Supplementary Fig. 11a).

To quantify the extent of admixture among lineages leading to the major groups of the *Podarcis* tree, we inferred reticulate phylogenetic networks of the species using phyloNet[21]. A total of 15 reticulations were identified by phyloNet, all of which were supported by introgression models with minimal errors from D-statistics (Supplementary Table 4, Supplementary Fig. 12). The results revealed extensive introgression and admixture throughout the evolution of *Podarcis* species (Fig. 3a, Supplementary

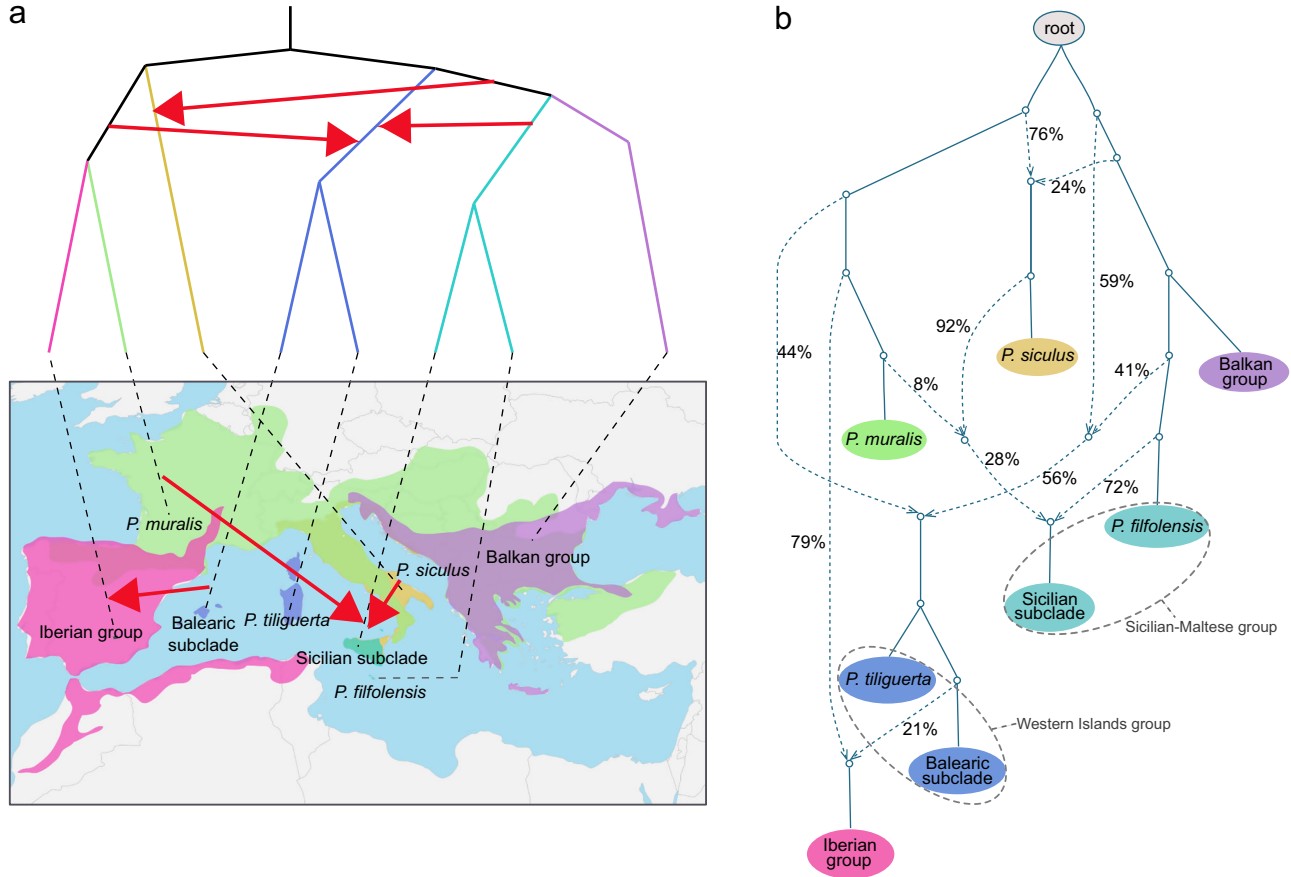

**Fig. 3 Extensive genetic exchange among major wall lizard lineages. a** A phylogram indicating the phylogenetic relationship among the six major groups of wall lizards and a map showing their geographic distributions. Note that the tree includes the splits within the Western Islands group and the Sicilian-Maltese group that are relevant for the origin of the mosaic genomes of island species. Red arrows indicate the introgression events. **b** The qpGraph of the major clades of wall lizards. The solid line represents the 'tree-like' evolution, whereas the dashed lines with arrows represent the parental ancestry of reticulations. The numbers next to dashed lines indicate the percentage of alleles derived from a given introgressive hybridization. For the corresponding illustration of genetic exchange within the major clades, see Supplementary Fig. 12.

Fig. 12). An estimation of the proportion of introgressed alleles from parental nodes in qpGraph[19] indicated that the reticulations involved 3–49% of alleles from the minority ancestry (Fig. 3b, Supplementary Fig. 12). In 4 out of 12 lineages that experienced introgression, the rate of protein-coding gene evolution (dN/dS ratios; a common estimate of selection), was higher for the introgressed genes than for genes with a history consistent with the consensus phylogeny (Supplementary Fig. 13a and Supplementary Table 5). These results are consistent with adaptive evolution of introgressed genes.

We further fitted admixture models for evolutionary scenarios containing subsets of species to validate the reticulation events using Nelder–Mead optimization[22]. The results revealed extensive, ancient reticulations between the major clades (Fig. 3). For example, the most recent common ancestor (MRCA) of the Western Islands group received 41% of alleles from the MRCA of the Sicilian-Maltese group, and 44% of alleles from the MRCA of the Iberian group and *P. muralis* (Fig. 3b). *P. siculus*, a species that is widely distributed across the Italian Peninsula and adjacent islands, received 24% of its alleles from the MRCA of the Balkan and the Sicilian-Maltese groups.

The phylogenetic network also suggested that introgression between evolutionarily younger lineages accounts for the discordances among local trees. Introgression was particularly prevalent for the Iberian group. First, three Iberian species (*P. carbonelli*, *P. guadarramae* and *P. vaucheri*) showed evidence of

multiple hybridization events between different lineages (Supplementary Fig. 12b). Second, the Iberian group hybridized with the *P. muralis* lineage inhabiting the Iberian Peninsula (Supplementary Fig. 12d). Similar patterns of extensive introgression were evident within the Balkan group (Supplementary Fig. 12c), with the phylogenetic network suggesting that *P. tauricus* has a mosaic genome with contributions from *P. gaigeae* (66% of alleles) and *P. ionicus* (34% of alleles). Similarly, the eastern lineage of *P. peloponnesiacus* has received 35% of its alleles from an extinct lineage.

**Mosaic genomes of Mediterranean island endemic species.** Across Mediterranean wall lizards, the hybridization events that resulted in a rather evenly shared ancestry were prevalent in the evolutionary history of extant island endemics. For the Western Islands group now occurring on Corsica, Sardinia and the Balearic islands, their ancestral lineage experienced introgression of 41% of alleles from the MRCA of the Sicilian-Maltese group, and 44% of alleles from the MRCA of *P. muralis* and the Iberian group. Following its separation from *P. tiliguerta*, the MRCA of the Balearic islands species provided 21% of the alleles into the Iberian group, before diversifying into the extant species (*P. lilfordi* and *P. pityusensis*; Fig. 3b). The admixture model based on D-statistics supported the inferred patterns of introgression with minimal error (M3: 122.51; versus M1: 13067.80, M2: 8281.35) by

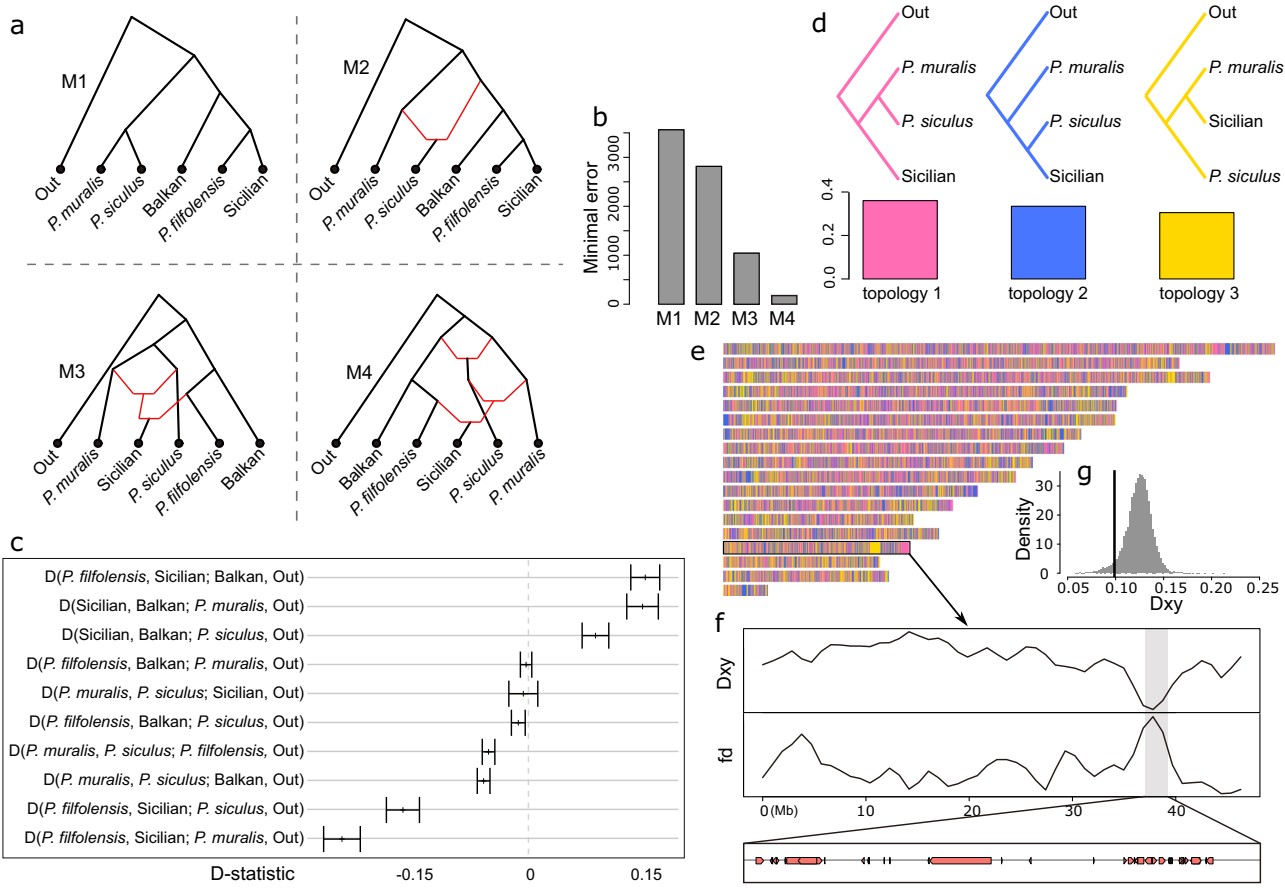

**Fig. 4 Reticulate evolution of wall lizards in the Sicilian-Maltese group. a** Proposed scenarios for the evolutionary history of wall lizards in Sicily and its surrounding area based on phyloNet networks: M1—consensus tree with no introgression; M2—an ancient introgression from the MRCA of Balkan and the two island groups into the *P. siculus* clade; M3—introgressions from *P. muralis* and *P. siculus* clades into the MRCA of the Sicilian subclade (*P. raffonei* and *P. waglerianus*); M4—all three reticulations combined. **b** The minimal errors for the four scenarios fitted by linear algebra and numerical optimization. **c** The observed D-statistic values for the species in the testing topologies. The dots represent the D-statistic scores, and the error bars represent their standard deviations. **d** Proportion of tree topologies supporting alternative relationships among *P. muralis*, *P. siculus* and the Sicilian subclade. **e** Distribution of tree topologies across the genome. Each colour concords with the topology in panel **d**. **f** 50 Kb window-based Dxy and fd statistics between *P. muralis* and the Sicilian subclade across Chromosome 15. The area shaded in grey marks a genomic block (36.3–39.5 Mb) of introgressed ancestry with low Dxy and high fd values containing 39 coding genes. **g** The genome-wide distribution of Dxy values between *P. muralis* and the Sicilian subclade based on 50 Kb window. The black line indicates the Dxy for the genomic block highlighted in panel **f**.

showing significant excess of allele sharing (Supplementary Fig. 14).

Similarly, two extant species in the Sicilian-Maltese group (*P. raffonei* and *P. waglerianus*), inhabiting Sicily and surrounding islands, descended from a lineage that received 28% of its alleles from *P. muralis* and *P. siculus*, with the latter descending from another hybrid lineage that shared 24% of its alleles with the MRCA of the Balkan and Sicilian-Maltese group (Figs. 3b and 4a). The D-statistics from the admixture model supported this complex scenario with three reticulation events (minimal error for M4: 175.60; versus M1: 3560.91, M2: 2813.70, M3: 1040.19; Fig. 4b, c).

Following hybridization, the introgressed genomic regions of parental species are bound to break down due to recombination, and regions that carry incompatible alleles will be purged[23]. Accordingly, introgressed loci are expected to be enriched in genomic regions with a high recombination rate and few genes[8,24,25]. Unusually long genomic blocks of different ancestry may therefore be putative candidates for adaptive introgression. One particularly important selective pressure during genome stabilization is mito-nuclear compatibility[26], suggesting that genomic regions that contain genes involved in cellular

respiration or energy metabolism should co-introgress with mitochondrial genomes.

An exceptionally well-suited case for testing this hypothesis is the two island endemic species *P. raffonei* and *P. waglerianus* (the Sicilian subclade in the Sicilian-Maltese group) that show evidence of mtDNA introgression from the *P. muralis* lineage (Supplementary Fig. 6). To look for evidence of co-introgression of the mitochondrial genome and genomic blocks, we scanned genomic windows for long runs of introgressed ancestry in these species (Supplementary Fig. 22). The longest region was a gene-dense region (36.3–39.5 Mb) on Chromosome 15, containing a total of 39 protein coding genes (Fig. 4f, Supplementary Table 6) that was shared with *P. muralis* (Fig. 4d, e). The genes residing in this introgressed genomic block included several genes (e.g. *ATP6V1*, *RAB11*) involved in energy metabolism and mitochondrial function. The finding that *P. muralis* and the Sicilian subclade formed sister taxa in the mitochondrial phylogeny (Supplementary Fig. 6) is consistent with a role of mito-nuclear compatibility in maintaining the introgressed genomic block. Furthermore, the average dN/dS ratios of the 39 genes residing in the introgressed genomic block was significantly higher than those of genes without a history of introgression (0.420 vs. 0.254;

permutation test: $P = 0.039$; Supplementary Fig. 13b and Supplementary Table 5). To further confirm that this genomic block is best explained by an introgression between *P. muralis* and the Sicilian subclade, we compared genome-wide patterns of Dxy and fd statistics (Fig. 4f). The Dxy of the candidate block (0.099) was lower than comparable values for 90% of the genome (Fig. 4g), implying that the origin of this genomic block was more recent than the divergence of the majority of their genomes. The fd of the candidate block (0.098) was significantly larger than genome-wide levels (0.038; permutation test: $P < 0.001$), further supporting the introgression of this genomic block between *P. muralis* and the Sicilian subclade.

## Discussion

The evolutionary history of wall lizards demonstrates that introgressive hybridization can be a persistent feature of animal clades over millions of years. While hybridization is well known to allow transfer of a limited portion of the genome between otherwise reproductively isolated lineages[1,27], the independent evolution and further diversification of animal lineages with major contributions of two or more taxa has been considered exceptional[2–5,27]. The rampant introgression throughout the diversification of wall lizards paints a different picture, and suggests that reticulated evolution may have played an important role in generating the exceptional diversity of the Mediterranean lizard fauna.

Climatic oscillations and changes in land masses have caused recurrent range contractions and expansions in the Mediterranean, promoting bouts of evolutionary divergence and secondary contact between lineages[28]. One particularly dramatic period in the history of the Mediterranean is the Messinian Salinity Crisis (~6.0–5.3 MYA), during which the Mediterranean Sea nearly desiccated before it was rapidly refilled to attain the approximate shape of today. While genomic data alone cannot provide conclusive evidence for the timing of diversification events[29], the diversifications within the two species-rich Balkan and Iberian groups of wall lizards appear to coincide with, or follow closely, this event. It could be predicted that such bursts in diversification would spring from hybridization events or that they would promote genetic exchange between incipient species[30,31]. However, our results demonstrate that introgression has been pervasive throughout the evolutionary history of wall lizards, even between distantly related lineages. Thus, the reticulated evolution of wall lizards is the result of occasional, but often extensive, genetic exchange between steadily evolving lineages, rather than a burst of hybridization associated with a rapid adaptive radiation.

Such a high and persistent level of introgression is possible because evolution of complete reproductive isolation is typically a slow process[11]. For example, following introductions outside of its native range by humans, the Italian wall lizard, *P. siculus*, has hybridized with species from the Balkan, Sicilian-Maltese and Western Islands groups, despite the lineages not having shared a common ancestor for over 17 million years[32,33]. At the same time, however, even closely related species can co-exist in sympatry[34], and the narrow hybrid zones between such species (e.g. within the Iberian group[34]) or subspecies (e.g. within the widely distributed *P. muralis*[35]) demonstrate that a few million years are typically sufficient to evolve pre- or post-copulatory mechanisms that prevent lineages from fully merging. It is, therefore, perhaps surprising that frequently more than 20%, and occasionally close to 50%, of the genomes of several extant species derive from hybridization between highly divergent lineages. This was particularly striking for the two species groups that occupy islands in the Tyrrhenian and Balearic seas, suggesting that Mediterranean islands have provided suitable conditions for the

formation of hybrid lineages, or that hybrid lineages have proven particularly able to persist in island habitats. Further studies of population genomic data in combination with reconstruction of the geological history could potentially identify the processes responsible for the origin and maintenance of these mosaic genomes[9,36].

Regardless of their origin, the evolutionary potential of lineages with extensive genomic admixture is evident from the fact that several of them went on to diversify into new species (e.g. *P. raffonei* and *P. waglerianus* in the Sicilian-Maltese group). Another striking feature of these species with mosaic genomes, in particular island endemics (e.g. *P. pityusensis* in the Western Islands group[37]), is their extraordinary phenotypic variability. Lizards from nearby islets are frequently more diverse and disparate in body size, shape and colouration than are different mainland species (Fig. 1), an observation that has puzzled naturalists and evolutionary theorists since the 19th century[12,13]. The mixed ancestry of these species provides a plausible explanation for their striking variability. The merging of genomes that have evolved independently for millions of years creates opportunities for novel phenotypes to emerge, and ample opportunity for drift and selection to cause populations to differentiate. Conversely, the extensive genomic introgression between lineages may have facilitated the long-term persistence of colour morphs that are shared between species[14,38]. Finally, the presence of long genomic blocks in otherwise highly recombined hybrid genomes suggests scope for genomic conflict and adaptive introgression to have shaped the genomic and phenotypic features of extant species. While further data is necessary to identify exactly how hybridization contributed to their evolvability and diversity, genetic exchange has been a pervasive feature of the evolution of Mediterranean lizards.

## Methods

**Sample information and sequencing**. A total of 36 samples were considered in this study, representing 26 species of the *Podarcis* genus, eight lineages within recognized species, and two species (*Atlantolacerta andreanskyi* and *Archaeolacerta bedriagae*) as outgroup (Supplementary Table 1). All specimens were collected in accordance with the policy of the animal care and use ethics of local institutions (for collection permits, see Supplementary Table 7). Genomic DNA was extracted using a DNeasy Blood & Tissue Kit (Qiagen, USA) according to manufacturer's instructions. Short-insert (300–500 bp) libraries were sequenced on an Illumina HiSeq X platform by NOVOGENE Ltd. (Hong Kong). The sequence reads were quality checked using FASTQC (http://www.bioinformatics.babraham.ac.uk/projects/fastqc/) and trimmed using trimmomatic[39] using default settings except for "LEADING:3, TRAILING:3, SLIDINGWINDOW:4:5, MINLEN:70". Clean reads were aligned to the *P. muralis* reference genome version PodMur_1.0[14] using bwa-mem (version 0.7.1; http://bio-bwa.sourceforge.net/bwa.shtml#2/). Single nucleotide and short indel variants were called using the GATK best practice workflow[40], with following cutoffs for filtering: SNP quality >100, base quality >30, mapping quality >50, quality by depth >2.0, minimum depth >288, and maximum depth <792, and other default parameters in GATK[40].

**Phylogenetic framework**. Two strategies were applied to obtain a robust *Podarcis* phylogeny. First, we used a concatenation approach by concatenating all SNVs across the whole genome sequences (WGS) and protein-coding sequences (CDS), respectively, and inferred the Maximum likelihood (ML) trees. Second, we adopted a multispecies coalescent approach to infer the *Podarcis* tree using ASTRAL-III[41]. The linkage disequilibria of the 34 wall lizard lineages are unknown, and we therefore considered several different window sizes. We generated the local tree dataset by separately splitting the whole genome into fixed windows of 200, 100, 50, 25, 10 and 5 kb, and reconstructed the ML trees for each window. Only windows with the missing rate of SNVs <1% were retained in the following analyses. ML trees were inferred using IQTree[42,43] with an GTR + ASC model and 1000 bootstrap replicates. The topologies of local ML trees (based on 50 kb fixed-windows) were quantified using *Twisst*[44] and plotted with different colours in R using the function plot (https://www.r-project.org/). We also used the Bayesian coalescent model in SNAPP from the BEAST2 package[45,46] to infer the *Podarcis* tree based on 3000 randomly selected variants due to computational constraints. Robinson–Foulds (RF) distances were calculated for each pair of phylogenetic topologies using the R package 'ape'[47] to assess the discrepancy among phylogenies.

**Mitochondrial genome**. The mitochondrial genome of each individual was de novo assembled using NOVOPlasty[48]. The mitochondrial genome of *P. muralis* (accession FJ460597 from MitoZoa[49]) was set as a starting reference. A total of 6 Gb sequence reads from each sample were randomly extracted for the baiting and iterative mapping with default parameters. For information on the completeness of the mitochondrial genome assemblies for each lineage, see Supplementary Table 3. Mitochondrial genomes were aligned using MUSCLE[50]. All ambiguous regions were excluded from the analyses to avoid false hypotheses of primary homology. IQTree was used to infer the phylogenetic tree (ML) with 1000 boostrap replicates after partition model selection[42,43].

**Divergence time estimation**. Given the extensive admixture during the evolutionary history of *Podarcis* species, we estimated the lineage divergence times based on genomic regions for which local trees were concordant with the consensus phylogeny (resulting in 1.24 million SNVs, 4.37% of the WGS dataset). A relaxed clock[45] model in MCMCtree from the package PAML[51] was used to estimate divergence times. For MCMCTree, the calibration constraints were specified with soft boundaries by using 0.025 tail probabilities above and below the limit in the built-in function of MCMCtree. The independent rate model (clock = 2) was used to specify the rate priors for internal nodes. The MCMC run was first executed for 10,000,000 generations as burn-in and then sampled every 150 generations until a total of 100,000 samples were collected. Two MCMC runs using random seeds were compared for convergence, and similar results were found. For reference, we also estimated the divergence times based on the entire WGS dataset and on mitochondrial DNA, which yielded qualitatively identical, and quantitatively very similar, results (Supplementary Fig. 23).

**Estimation of species diversification and demographic history**. We estimated the diversification rate through time using a Lineage Through Time (LTT) curve and a sliding-window estimation[18]. Shifts in diversification rates were inferred using ML utilizing treePar[52] and Bayesian inference using RevBayes[53]. Pairwise sequentially Markovian coalescence (PSMC) analysis[54] was applied to estimate the demographic history of each genome with the command "psmc -N25 -t15 -r5 -p4 + 25*2 + 4 + 6" (Supplementary Figs. 24–27). Mutation rates were estimated by r8s[55], and the generation time was set to two years.

**Admixture and introgression analysis**. We first applied the standard ABBA-BABA test (Patterson's D-statistics) using the qpDstat command in AdmixTools[19], and considered all triplets of the WGS tree, using *A. bedriagae* as the outgroup. We assessed significance through a block-jackknifing approach as implemented in AdmixTools, and applied a Bonferroni correction to assign significance at the 95% confidence level. In addition, we phased the genomes for each sample using BEAGLE 4.1[56] with a uniform recombination map and options "-x 1000000 -y 200000 -z 1000" and then used the chromopainter software in the fineSTRUCTURE package[20] to calculate the 'co-ancestry matrix'—a summary of nearest-neighbour haplotype relationships that is an indication of admixture and introgression.

We conducted phylogenetic network analyses using phyloNet[21] to infer reticulation events among species. Due to computational limitations, we were unable to analyze the whole dataset at once. We therefore divided the samples into different groups, including (1) the major clades, (2) the Balkan group, (3) the Iberian group, and (4) *P. muralis* together with Iberian species. We made use of high-quality local trees with mean bootstrap >80, extracted 2000 random trees per run with a chain-length of 10,000,000 and a burn-in of 5,000,000 in the MCMC_gt module. We used 100 independent iterations for each run, and extracted all output networks with more than 50% posterior probability, and summarized the results by generating a correlation matrix of those networks based on Luay Nakhleh's metric of reduced phylogenetic network similarity[8]. In addition, we also used the Infer_Network_MPL module to infer reticulation events based on maximum pseudo-likelihood by setting maximum reticulations of 5 for 50 iterations.

To validate the phylogenetic networks, we further fitted the evolutionary scenarios by admixture models for subsets of species or lineages related to introgression events using a combination of linear algebra and numerical optimization (Nelder–Mead) based on observed D-statistics[19]. The minimal error $(F - f)^t \times S^{-1} \times (F - f)$ is defined as the cost function of the model parameters, where $F$ and $f$ are the expected and observed D statistics, respectively, and $S$ is the covariance matrix of $f$. The consensus tree topology was always fitted first, then the other inferred reticulations were gradually added. The best-fitting scenario was identified as the one with the minimum error. The analysis was performed using the R package 'Admixgraph'[22].

Based on the phylogenetic network, we used the program qpGraph from AdmixTools[19] to fit the evolutionary history for all 34 *Podarcis* species or lineages together with introgressions. qpGraph optimizes the fit of a proposed admixture graph in which each node can be descended either from a mixture of two other nodes, or from a single ancestral node. The proportion of introgressed alleles was calculated by $f4$-ratio tests.

The method quantifying introgression via branch lengths (QuIBL)[8], based on the counts of triplet topologies of local ML trees, was used to quantify the frequency of introgression for species or lineages. For each set of trees we determined the likelihood that the branch lengths were best described by a simple

exponential distribution as expected under ILS or a mixture of ILS and either introgression or speciation processes. The Bayesian information criterion (BIC) was used to identify the best-fitting model.

To confirm introgression of candidate genomic regions, we calculated the absolute genetic divergence (Dxy) between donor and recipient lineages, and the fd statistics with the same triplets as used for the calculation of D statistics based on 50 Kb fixed-window genomic regions. Dxy and fd were calculated using the genomics_general package (https://github.com/simonhmartin/genomics_general/), where a significantly lower Dxy and higher fd support introgression[57] (tested by 1000 permutations).

**Evolution of protein-coding genes**. Signatures of selection were estimated by the ratios between non-synonymous and synonymous substitution rate (dN/dS) for protein-coding genes in codeml from the package PAML[51] using runmode '-2'. For each recipient lineage with a history of introgression, we identified the coding sequences of genes from genomic regions derived from introgression events (foreground), and genes with a history consistent with the consensus phylogeny (background). We compared substitution rates between each of the 12 focal lineages that experienced a total of 15 introgression events (Supplementary Table 4) compared to the outgroup *Archaeolacerta bedriagae*. To test if the dN/dS ratios were significantly different between fore- and background genes, we used 1000 permutations.

**Reporting summary**. Further information on research design is available in the Nature Research Reporting Summary linked to this article.

## Data availability

All sequence data generated in this study have been deposited in NCBI Sequence Reads Archive (SRA) with accession number PRJNA715201.

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

## Acknowledgements

A.K. thanks the FEDER Funds through the Operational Competitiveness Factors Program—COMPETE and National Funds through FCT—Foundation for Science and Technology within the scope of the project "PTDC/BIA- EVL/28090/2017-POCI-01-0145-FEDER-028090" for financial support in the collection of samples. D.S. thanks the Italian Ministry of Education, University and Research for financial support (PRIN project 2017KLZ3MA). We thank the following people for support in the collection of specimens: A. Castilla, A. Perera, B. Santos, B. Tomé, C. Rato, D. Rosado, E. Garcia-Muñoz, F. Jorge, G. Caeiro-Dias, G. Pérez i de Lanuza, I. Damas, I. Rocha, I. Tavares, J. Santos, M. Ribeiro, M. A. Carretero and V. Gomes. T.U. thanks the Knut and Alice Wallenberg foundation for support through a Wallenberg Academy fellowship, and project support from the Swedish Research Council (2014_04465 and 2017_03846) and the Crafoord Foundation (20160911 and 20190784).

## Author contributions

W.Y., N.F. and T.U. designed and coordinated the study. A.K., D.J.H., G.M.W., D.S. and T.U. performed fieldwork. W.Y. performed DNA extractions. W.Y. performed genomic analysis with advice from N.F. and T.U. W.Y., N.F., C.P., D.S. and T.U interpreted the results. T.U. wrote the manuscript with assistance from W.Y. and N.F., with comments from, and final version approved by, all authors.

## Funding

## Competing interests

The authors declare no competing interests.
