## [Peer Review File · Nature Communications]

Reviewers' Comments:

Reviewer #1:

Remarks to the Author:

This manuscript sheds light on the evolutionary history of Mediterranean wall lizards with an impressive dataset of full genome sequencing of 26 species. The authors find extensive gene exchange throughout the phylogeny and time. They have taken full advantage of the newest and best methods to investigate hybridization in the evolutionary history of taxa in a phylogenetic context. The combination of these methods shows a clear picture of rampant admixture that affected most wall lizard species at some point in their ancestry. The authors aim to study if 'introgressive hybridization has contributed to the exceptional levels of biodiversity and endemism of the Mediterranean fauna' and 'identify drivers of diversification.' However, while their study shows convincingly that introgressive hybridization is pervasive, they do not test if hybridization promoted biodiversity. Focusing on genomic regions with strong signatures of introgression could be one approach, or testing if genes showing signatures of adaptation are more often introgressed than the genome-wide background, or testing if taxa with more evidence of hybridization show greater genetic and phenotypic diversity.

Given the rampant admixture, it is not ideal though to construct dated nuclear genome trees as a dichotomous tree model is a poor model for such a reticulate evolutionary history. The comparison between panels a and c in Fig. 2 makes that quite apparent. As an example, in the time-calibrated ML tree, the divergence between *P. muralis* lineages is estimated to have occurred at the same time as that of most Iberian group lizards. However, in the Densitree in panel c suggests that the *P. muralis* lineages are way more closely related than the Iberian group species. Another example would be *P. siculus* which in panel a is estimated to have diverged in the distant past from *P. muralis* and the Iberian group. However, as can be seen in panel c, the divergence was in fact much more recent but the split is overestimated in the concatenated tree as *P. siculus* has received a large part of its ancestry from the Balkan/Sicilian-Maltese group lizards and relatives. Inferring diversification rates from a dichotomous tree is thus not appropriate for such a reticulate history. It would be better to use the mitochondrial tree for dating (with the caveat of cytonuclear discordance) or identify genomic regions that show little evidence of hybridization and use those.

More detail is needed for the Methods: e.g. What parameters were used for Trimmomatic? How was dxy and fd computed? What window size was used for these? Finestructure requires phasing. How was the data phased? How was the genomic region identified that is supposedly co-introgressed with mtDNA (L. 209)? It seems that TWISST was used or what are the tree topologies support in panel d and e. There is no mentioning of these in the Methods. Or was Chromopainter used? Unclear.

For the expected correlation between gene flow and recombination Schumer et al. 2018, Science, should be cited. It would be good if the authors could check if in general this expectation is met in the wall lizards. It might even be possible to check if the correlation becomes stronger the more divergent the hybridising lineages are. This would be strong support for the accumulation of incompatibilities over time that the authors mention.

Minor comments:

L. 37: Here it would be good to cite the recent review on the topic of hybridization and adaptive radiation by Marques et al. 2019, Trends in Ecology and Evolution, and also a paper on plants, e.g. sunflowers. A lot of research on hybridization was done on plants and it would be good to cite some of them.

Fig. 2: The alignment of boxes indicating how the nodes correspond to wall lizard groups seems to be shifted. e.g. is the Sicilian-Maltese group really split between two highly divergent groups and what

is the bottom-most taxon above the outgroup?

Fig. 4: Unclear how the tree topologies in panel e were generated. Are these sliding window trees, TWISST analyses, Chromopainter?

Extended Data Fig. 4: This is a key result in my opinion. To make it easier to read it would be good to add the colours of the different groups used in the main figures. Someone not familiar with the different species names will find it difficult to understand if the cytonuclear discordance supports the inferred introgression events.

Extended Data Fig. 8: Some of the labels are very tiny and hard to read.

L. 295: There was a recent review on this in Trends in Ecology and Evolution, Jamie and Meier, 2020. Maybe worth citing here. It would be good if the authors could test if hybridization indeed «extensive genomic introgression between lineages may have facilitated the long-term persistence of colour morphs that are shared between species». Given that some of the genes underlying colour morphs are known, it should be possible to test if these genes show more discordance from the species tree than other genes.

Reviewer #2:

Remarks to the Author:

This paper reports analysis of genome variation across the diverse radiation of Mediterranean wall lizards, *Podarcis*. Based on very substantial new data (34 new genome assemblies) and a series of sophisticated phylogenetic and population genomic analyses, the authors infer that introgression across species has occurred throughout the history of this radiation, and perhaps contributed to the unusually high phenotypic diversity shown by some island endemic species. They also elucidate one possible example of adaptive introgression involving cyto-nuclear interactions.

In general, this is a superb paper and one worthy of publication in Nature Comms. The system is fascinating and these new genomic data and analyses greatly extend previous observations of cyto-nuclear discordance to establish this as an exemplar of introgression-associated radiation. The data and methods are well described and results are carefully interpreted.

I have just a few comments and suggestions:

- Given the evidence of rampant introgression throughout the history of the radiation, what maintains species boundaries now? In Discussion, authors refer to some examples where recent introductions (e.g. *P. siculus*) have results in interspecific hybridisation and others (e.g. among paratric lineage of *P. muralis*) where there are narrow hybrid zones. Are there still other situations in this radiation where there is sympatry among species with not hybridisation or introgression?
- The examples of strong discordance between mtDNA and nuclear gene trees (lines 101-103) could be explained in more detail, esp as this underpins later test for co-introgression of nuclear genes relating to mtDNA function.
- I did worry a bit about reference bias in using the muralis genome to call SNPs in other taxa - some evidently with MRCA back to ~20Myr. How does missing SNV rates up to 1% (Suppl Fig 1) compare to average nucleotide divergence and could this somehow affect the D statistics etc. used here [maybe not, but I just can't intuit the outcomes].
- Please provide more information on the secondary calibration of divergence times in the phylogeny (ref 16).
- How might reticulation early in the radiation (ie towards the base of the tree) and later affect estimates of splitting times, and hence LTT plots and estimates of shifts in diversification rates? We know that methods such as concatenation-ML and ASTRAL are affected by introgression, so it would be as well to add appropriate caveats here.

- My understanding is that both PhyloNET and fineSTRUCTURE chromopainter need phased haplotype data, not diplotypes. If so, how did you meet this requirement?

Reviewer #3:

Remarks to the Author:

Using 34 whole genomes from 34 major lineages and 26 species the authors examine the evolutionary history of the enigmatic Mediterranean wall lizards. I really enjoyed reading this paper. The writing is clear, and the analyses are appropriate for examining the evolutionary history of the group. I think the authors could be a bit more careful with how they discuss the importance of hybridization in the evolutionary history of these wall lizards (see below), but I do think the data presented are compelling.

I rarely have so few comments on a paper, but I think this is already in great shape and there are no additional analyses that I would suggest the authors conduct.

Other comments:

title: hybrid origins - maybe tone this down given that you have not demonstrated hybrid origins per the Schumer et al. criteria. Perhaps "Widespread introgression and a history of hybridization in endemic Mediterranean lizards", or something like that?

Line 28 - 30 - I appreciate this suggestion, just be cautious since you haven't identified causal variants that, when combined in these lineages with extensive hybridization, might influence phenotype.

Line 32 - 34 - I would specify "in vertebrates" given that hybridization is not rare at all in plants

Line 36 - I think being careful with wording is important here. Hybrid lineage implies hybrid speciation, which is likely rare per Schumer et al. Instead, I think it would make more sense to refer to these lineages as ones within which extensive hybridization has happened. This may sound like semantics, but hybrid speciation implies a certain process and that hybridization itself led to reproductive isolation. This is like rare in general, as you note. The evidence for hybridization having occurred at some point during the evolutionary history of a given lineage, however, is growing and it seems like hybridization has played a role in the evolutionary history of many species.

Line 43 - 44 - I am not familiar with the flora and fauna of this region, but it would surprise me if no papers exist that show hybridization in the evolutionary history of a lineage.

Line 80 - Why use SNV when SNP is so much more widespread in the literature?

Line 200 - 201 - or structural rearrangements, right?

Fig 4 - some font is too small to read

Line 236 - panel d, right?

Below is the report of our manuscript revision. We show sentences taken from the manuscript in italic, and parts newly inserted into the manuscript as underlined. Page and line numbers refer to positions in the document with tracked changes.

REVIEWER COMMENTS

Reviewer #1 (Remarks to the Author):

This manuscript sheds light on the evolutionary history of Mediterranean wall lizards with an impressive dataset of full genome sequencing of 26 species. The authors find extensive gene exchange throughout the phylogeny and time. They have taken full advantage of the newest and best methods to investigate hybridization in the evolutionary history of taxa in a phylogenetic context. The combination of these methods shows a clear picture of rampant admixture that affected most wall lizard species at some point in their ancestry. The authors aim to study if ‘introgressive hybridization has contributed to the exceptional levels of biodiversity and endemism of the Mediterranean fauna’ and ‘identify drivers of diversification.’ However, while their study shows convincingly that introgressive hybridization is pervasive, they do not test if hybridization promoted biodiversity. Focusing on genomic regions with strong signatures of introgression could be one approach, or testing if genes showing signatures of adaptation are more often introgressed than the genome-wide background, or testing if taxa with more evidence of hybridization show greater genetic and phenotypic diversity.

Response: We thank the reviewer for the succinct summary of our manuscript and for the overall positive evaluation. Regarding the suggestion of attempting to causally link introgression with adaptive evolution, we remain cautious. This would be extremely challenging, if not impossible, from our data. We have accordingly modified the sentence in the introduction that wrongly gave the impression that this was the scope of our contribution, and also reversed the order of panels in Fig. 1 preventing the false impression that establishing the causes of the extraordinary diversity was our aim.

Originally: *Here, we use whole-genome sequences of representatives of all major lineages of the Podarcis genus to unravel their entangled evolutionary history and identify drivers of diversification.*

Revised (Abstract, lines 50-52): *Here, we use whole-genome sequences of representatives of all major lineages of the Podarcis genus to unravel their entangled evolutionary history.*

Nevertheless, the reviewer is correct that we could do a little more to substantiate that introgression not only is rampant, but also contributed to adaptive evolution. We have therefore added analyses that test if patterns of protein-coding gene evolution are consistent with the notion that introgressed genes have been under selection. To this end, we have estimated the ratios between non-synonymous and synonymous substitution rates (dN/dS) of protein-coding genes and contrasted introgressed genes with those that have an evolutionary history consistent with the consensus phylogeny. The results show that in a substantial number of the lineages that have mosaic genes, the introgressed genes show an elevated dN/dS ratio and are therefore consistent with adaptive evolution of introgressed genes. In particular, elevated dN/dS ratios are

pronounced in the large genomic block that co-introgressed together with mtDNA from *P. muralis* into the Sicilian subclade, further substantiating an adaptive significance of the candidate region. We have added these new insights into the Results section, modified the Methods section accordingly and added a new Supplementary Figure (6) and Table (4) summarizing these results.

Sentence added (Results, lines 162-166): *In 4 out of 12 lineages that experienced introgression, the rate of protein-coding gene evolution (dN/dS ratios; a common estimate of selection), was higher for the introgressed genes than for genes with a history consistent with the consensus phylogeny (Supplementary Fig. 6a and Supplementary Table 4). These results are consistent with adaptive evolution of introgressed genes.*

Sentence added (Results, lines 235-238): *Furthermore, the average dN/dS ratios of the 39 genes residing in the introgressed genomic block was significantly higher than those of genes without a history of introgression (0.420 vs. 0.254; permutation test: $P = 0.039$; Supplementary Fig. 6b and Supplementary Table 4).*

Paragraph added (Methods, lines 537-546): *Evolution of protein-coding genes* Signatures of selection were estimated by the ratios between non-synonymous and synonymous substitution rate (dN/dS) for protein-coding genes in codeml from the package PAML⁵³ using runmode '-2'. For each recipient lineage with a history of introgression, we identified the coding sequences of genes from genomic regions derived from introgression events (foreground), and genes with a history consistent with the consensus phylogeny (background). We compared substitution rates between each of the 12 focal lineages that experienced a total of 15 introgression events (Extended Data Table 1) compared to the outgroup *Archaeolacerta bedriagae*. To test if the dN/dS ratios were significantly different between fore- and background genes, we used 1,000 permutations.

Given the rampant admixture, it is not ideal though to construct dated nuclear genome trees as a dichotomous tree model is a poor model for such a reticulate evolutionary history. The comparison between panels a and c in Fig. 2 makes that quite apparent. As an example, in the time-calibrated ML tree, the divergence between *P. muralis* lineages is estimated to have occurred at the same time as that of most Iberian group lizards. However, in the Densitree in panel c suggests that the *P. muralis* lineages are way more closely related than the Iberian group species. Another example would be *P. siculus* which in panel a is estimated to have diverged in the distant past from *P. muralis* and the Iberian group. However, as can be seen in panel c, the divergence was in fact much more recent but the split is overestimated in the concatenated tree as *P. siculus* has received a large part of its ancestry from the Balkan/Sicilian-Maltese group lizards and relatives. Inferring diversification rates from a dichotomous tree is thus not appropriate for such a reticulate history. It would be better to use the mitochondrial tree for dating (with the caveat of cytonuclear discordance) or identify genomic regions that show little evidence of hybridization and use those.

Response: Thank you for raising this important point. Unfortunately, the reviewer was misled about divergence times from the Densitree in Fig. 2c, since the presentation of the tree omitted branch length. To avoid this problem, we have replaced the original tree with a Densitree that preserves branch length information and is therefore more directly comparable to the dated ML tree in Fig. 2a.

Information added to Fig. 2 legend (lines 137-139): *A ‘DensiTree’ illustration of the prevalent discordances among 500 randomly selected local trees derived from 200 kb windows (with branch length proportional to genetic change).*

Concerning the more general comment on the difficulty of estimating divergence time in a group with rampant introgression, such as wall lizards, we fully agree with the concern raised by the reviewer. To address this concern, we have added an additional analysis whereby we perform the dating on a dataset that contains only genomic regions for which tree topologies are consistent with the consensus tree. Although this analysis involved only 4.4% of the total dataset (a result of the extensive introgression across the phylogeny), the results were consistent with this original analysis performed on the entire WGS dataset. In the revised manuscript, we present the results on the selected genomic regions in the main part and replaced Extended Data Fig. 5 with these new results. The original result, together with a divergence time estimation based on mitochondrial DNA, is now presented in the Supplementary material (Supplementary Fig. 8).

Sentence added (Results, lines 115, 116): *To reduce the influence of extensive introgression, we used only the genomic regions for which local trees were concordant with the consensus phylogeny.*

Originally: *We estimated that the split among the major clades of Podarcis wall lizards took place in the Miocene between ~17.6-11.8 MYA (Fig. 2, Extended Data Fig. 5).*

Revised (Results, lines 118-120): *We estimated that the split among the major clades of Podarcis wall lizards took place in the Miocene between ~16.7-9.8 MYA (Fig. 2, Extended Data Fig. 5).*

Sentences added (Methods, lines 458-475): *Given the extensive admixture during the evolutionary history of Podarcis species, we estimated the lineage divergence times based on genomic regions for which local trees were concordant with the consensus phylogeny (resulting in 1.24 million SNVs, 4.37% of the WGS dataset). [...] For reference, we also estimated the divergence times based on the entire WGS dataset, which yielded qualitatively identical, and quantitatively very similar, results (Supplementary Fig. 8).*

Sentence added (Legend of Extended Data Fig. 5, lines 645-647): *The results shown in panels c to g are based on selected genomic regions where local trees were concordant with the consensus phylogeny.*

More detail is needed for the Methods: e.g. What parameters were used for Trimmomatic? How was dxy and fd computed? What window size was used for these? Finestructure requires phasing. How was the data phased? How was the genomic region identified that is supposedly co-introgressed with mtDNA (L. 209)? It seems that TWISST was used or what are the tree topologies support in panel d and e. There is no mentioning of these in the Methods. Or was Chromopainter used? Unclear.

Response: We agree with the reviewer and have added more information to our Methods section. We have now added the following information to address each of the above points.

Information added (Methods, lines 419-421): *[...] trimmed using trimmomatic⁴⁰ using default settings except for “LEADING:3, TRAILING:3, SLIDINGWINDOW:4:5, MINLEN:70”.*

Information added (Methods, lines 491-494): *In addition, we phased the genomes for each sample using BEAGLE 4.1⁵⁷ with a uniform recombination map and options “-x*

1000000 -y 200000 -z 1000" and then used the chromopainter software in the fineSTRUCTURE package²⁰ to calculate the 'co-ancestry matrix' [...].

Sentence added (Methods, lines 438-404): The topologies of local ML trees (based on 50 kb fixed-windows) were quantified using Twisst⁴⁵ and plotted with different colors in R using the function plot (<https://www.r-project.org/>).

Originally: *To identify the genomic regions with signatures of introgression, we calculated the absolute genetic divergence (Dxy) between species or lineage pairs, and the fd statistics based on 50 Kb windows across the genome.*

Revised (Methods, lines 528-532): To confirm introgression of candidate genomic regions, we calculated the absolute genetic divergence (Dxy) between donor and recipient lineages, and the fd statistics with the same triplets as used for the calculation of D statistics based on 50 Kb fixed-window genomic regions.

Originally: *A significantly lower Dxy and higher fd identify an introgressed genomic region⁵⁸ tested by 1,000 permutations.*

Revised (Methods, lines 532-535): Dxy and fd were calculated using the genomics general package ([https://github.com/simonhmartin/genomics general/](https://github.com/simonhmartin/genomics_general/)), where a significantly lower Dxy and higher fd support introgression⁵⁸ (tested by 1,000 permutations).

For the expected correlation between gene flow and recombination Schumer et al. 2018, Science, should be cited. It would be good if the authors could check if in general this expectation is met in the wall lizards. It might even be possible to check if the correlation becomes stronger the more divergent the hybridising lineages are. This would be strong support for the accumulation of incompatibilities over time that the authors mention.

Response: We thank the reviewer for this suggestion. We agree that it would be very informative to investigate if introgressed regions are particularly common in regions of the genome with high recombination rates. We added reference to Schumer *et al.* who has demonstrated this pattern in swordtail fish (lines 217, 218) and also Edelman et al. 2019 and Nelson et al. 2020 who documented this pattern for *Heliconius* butterflies and monkeyflowers, respectively. To perform the equivalent analyses in wall lizards, a fine-scale recombination map would be required. Unfortunately, this necessitates extensive population genomic data. Thus, our current data is insufficient to robustly address this question, but it would be an interesting target for future studies.

Minor comments:

L. 37: Here it would be good to cite the recent review on the topic of hybridization and adaptive radiation by Marques et al. 2019, Trends in Ecology and Evolution, and also a paper on plants, e.g. sunflowers. A lot of research on hybridization was done on plants and it would be good to cite some of them.

Response: We thank the reviewer for this suggestion and have added reference to Marques *et al.*, 2019, TREE (lines 38, 39). However, to keep our main focus on animals (see comment by reviewer #3 below), we refrain from citing plant adaptive radiations in this context.

Fig. 2: The alignment of boxes indicating how the nodes correspond to wall lizard groups seems to be shifted. e.g. is the Sicilian-Maltese group really split between two highly divergent groups and what is the bottom-most taxon above the outgroup?

Response: We thank the reviewer for picking up this error in our illustration. We have now modified Fig. 2c and made sure that the labelling is unambiguous.

Fig. 4: Unclear how the tree topologies in panel e were generated. Are these sliding window trees, TWISST analyses, Chrompainter?

Response: We apologize for not providing sufficient information. Different tree topologies were quantified using TWISST. The local trees were estimated based on 50 Kb fixed-window genomic regions. The details have been added in lines 438-440 (see also response above).

Extended Data Fig. 4: This is a key result in my opinion. To make it easier to read it would be good to add the colours of the different groups used in the main figures. Someone not familiar with the different species names will find it difficult to understand if the cytonuclear discordance supports the inferred introgression events.

Response: We agree with the reviewer and have modified the figure to make it visually more appealing and more intuitive to grasp.

Extended Data Fig. 8: Some of the labels are very tiny and hard to read.

Response: We have reworked all figures to avoid font sizes that are difficult to read. Thank you for this advice.

L. 295: There was a recent review on this in Trends in Ecology and Evolution, Jamie and Meier, 2020. Maybe worth citing here. It would be good if the authors could test if hybridization indeed «extensive genomic introgression between lineages may have facilitated the long-term persistence of colour morphs that are shared between species». Given that some of the genes underlying colour morphs are known, it should be possible to test if these genes show more discordance from the species tree than other genes.

Response: We agree that this review is relevant and have added a reference to it (lines 419, 420). Regarding the suggestion of singling out candidate genes underlying color polymorphisms in wall lizards and scrutinizing their evolutionary history, we agree that it would be a promising line of research. However, we believe that the current data is insufficient to gain any insight into this question since there are too many unknown variables. For example, there is not sufficiently robust data on the distribution of color polymorphisms across *Podarcis*, and addressing this issue fully would require population-level genomic data. We have therefore reserved this question for future research.

Reviewer #2 (Remarks to the Author):

This paper reports analysis of genome variation across the diverse radiation of Mediterranean wall lizards, *Podarcis*. Based on very substantial new data (34 new genome assemblies) and a series of sophisticated phylogenetic and population genomic analyses, the authors infer that introgression across species has occurred throughout the history of this radiation, and perhaps contributed to the unusually high phenotypic diversity shown by some island endemic species. They also elucidate one possible example of adaptive introgression involving cyto-nuclear interactions.

In general, this is a superb paper and one worthy of publication in Nature Comms. The system is fascinating and these new genomic data and analyses greatly extend previous observations of cyto-nuclear discordance to establish this as an exemplar of introgression-associated radiation. The data and methods are well described and results are carefully interpreted.

Response: We are grateful for the strong endorsement and appreciation of our study.

I have just a few comments and suggestions:

- Given the evidence of rampant introgression throughout the history of the radiation, what maintains species boundaries now? In Discussion, authors refer to some examples where recent introductions (e.g. *P. siculus*) have resulted in interspecific hybridisation and others (e.g. among paratric lineage of *P. muralis*) where there are narrow hybrid zones. Are there still other situations in this radiation where there is sympatry among species with not hybridisation or introgression?

Response: The reviewer raises an interesting point, and we can confirm that *Podarcis* species can indeed co-occur in sympatry without apparent gene flow between them. We have added this information.

Originally: *At the same time, however, the narrow hybrid zones between closely related species (e.g., within the Iberian group³⁵) or subspecies (e.g., within the widely distributed *P. muralis*³⁶) demonstrate that a few million years are typically sufficient to evolve pre- or post-copulatory mechanisms that prevent lineages from fully merging.*

Revised (Discussion, lines 294-299): *At the same time, however, even closely related species can co-exist in sympatry³⁵, and the narrow hybrid zones between such species (e.g., within the Iberian group³⁵) or subspecies (e.g., within the widely distributed *P. muralis*³⁶) demonstrate that a few million years are typically sufficient to evolve pre- or post-copulatory mechanisms that prevent lineages from fully merging.*

- The examples of strong discordance between mtDNA and nuclear gene trees (lines 101-103) could be explained in more detail, esp as this underpins later test for co-introgression of nuclear genes relating to mtDNA function.

Response: The importance of the discordance between the mtDNA and nuclear gene trees was also pointed out by Reviewer #1. To give these concordances more prominence in the manuscript, we have added further detail on the mtDNA introgressions to the Result section.

Sentence added (Results, lines 109-112): *These discordances between nuclear and mitochondrial DNA indicate introgression of mtDNA from geographically adjacent,*

but sometimes distantly related, donor lineages into the *P. hispanicus* 'GAL' lineage, into the *P. siculus* lineage, into the *P. tiliguerta* lineage and into the Sicilian subclade.

- I did worry a bit about reference bias in using the muralis genome to call SNPs in other taxa - some evidently with MRCA back to ~20Myr. How does missing SNV rates up to 1% (Suppl Fig 1) compare to average nucleotide divergence and could this somehow affect the D statistics etc. used here [maybe not, but I just can't intuit the outcomes).

Response: We agree that this is an important question. Since the genome sequence of *P. muralis* was used as a reference in this study, the genotyping rate was decreasing for species relatively distant to *P. muralis* (and therefore with higher nucleotide divergence). However, the average missing rate was low across all *Podarcis* lineages (0.08% - 1.16%; average 0.56%; shown in Supplementary Fig. 1). As is customary in phylogenomics, all window-based analyses (local tree inference, phyloNet analysis, candidate region identification including Dxy and fd statistics) were performed after imposing a cut-off on the maximum missing rate of SNVs (1%) per window. This ensures that only windows with sufficient signal are considered. Apart from technical errors, missing SNV could be pronounced in regions of the genome with an exceptionally fast evolutionary rate. While this is a limitation of every comparative genomics study, we find it unlikely that missing SNVs were biasing our results. Most importantly, the fact that we recovered comparable results with single-SNV-based and window-based analyses suggests that our results are not sensitive towards excluding low-quality windows. We have added information on the cut-off to the manuscript.

Sentence added (Methods, lines 436, 437): Only windows with the missing rate of SNVs < 1% were retained in the following analyses.

- Please provide more information on the secondary calibration of divergence times in the phylogeny (ref 16).

Response: This information was provided in the Methods section of the original version of the manuscript, but we have now moved it to the Results section for clarity.

Sentence moved (lines 116-118): *Two secondary calibrations were adopted from a recent phylogeny of the Lacertidae¹⁷, namely the root node (37.55 million years ago, MYA) and the crown node of Podarcis (18.60 MYA).*

- How might reticulation early in the radiation (ie towards the base of the tree) and later affect estimates of splitting times, and hence LTT plots and estimates of shifts in diversification rates? We know that methods such as concatenation-ML and ASTRAL are affected by introgression, so it would be as well to add appropriate caveats here.

Response: A similar concern was raised by Reviewer #1 and we fully agree that reticulations may bias estimates of divergence times and diversification rates. As explained in detail above, we have confirmed the dating on a dataset that contains only genomic regions for which tree topologies

are consistent with the consensus tree, and report this analysis in the main text. For more details, please see the reply to Reviewer #1 above.

- My understanding is that both PhyloNET and fineSTRUCTURE chromopainter need phased haplotype data, not diplotypes. If so, how did you meet this requirement?

Response: The reviewer is correct in that the fineSTRUCTURE chromopainter indeed requires phased genomic data. We used the software BEAGLE to obtain this data. For PhyloNET, we used the MCMC_gt module that requires local trees instead of genome data in the analysis. This information was added to the Methods (lines 491-493).

Reviewer #3 (Remarks to the Author):

Using 34 whole genomes from 34 major lineages and 26 species the authors examine the evolutionary history of the enigmatic Mediterranean wall lizards. I really enjoyed reading this paper. The writing is clear, and the analyses are appropriate for examining the evolutionary history of the group. I think the authors could be a bit more careful with how they discuss the importance of hybridization in the evolutionary history of these wall lizards (see below), but I do think the data presented are compelling.

Response: We thank the reviewer for the overall positive evaluation and for the constructive feedback. Concerning our interpretation of the importance of the hybridization for the evolutionary history of wall lizards, we agree that this cannot be conclusively established from our data. Reviewer #1 raised a similar concern and we have accordingly removed the statement about 'identifying drivers of adaptation' from our manuscript (line 52).

I rarely have so few comments on a paper, but I think this is already in great shape and there are no additional analyses that I would suggest the authors conduct.

Response: We are very glad about this appreciation of our work.

Other comments:

title: hybrid origins - maybe tone this down given that you have not demonstrated hybrid origins per the Schumer et al. criteria. Perhaps "Widespread introgression and a history of hybridization in endemic Mediterranean lizards", or something like that?

Response: We thank the reviewer for this suggestion. We take that the reviewer has issues with using the term 'hybrid origin' in this context because s/he understands this term as implying that hybridization caused reproductive isolation. The extent to which hybridization causes reproductive isolation is a conceptually important point but, by 'origin', we simply mean that the ancestors of contemporary lineages came from (at least) two different lineages. However, we acknowledge that there is room for misunderstanding since the title does not provide any additional context on how we use this term. We have therefore revised the title.

Originally: *Rampant introgression and hybrid origins of endemic Mediterranean lizards*

Revised (Title): *Rampant introgression and mosaic genomes of Mediterranean endemic lizards*

Line 28 - 30 - I appreciate this suggestion, just be cautious since you haven't identified causal variants that, when combined in these lineages with extensive hybridization, might influence phenotype.

Response: Agreed, as explained above, we have used this opportunity to reword this sentence to not overstate what the data can deliver.

Line 32 - 34 - I would specify "in vertebrates" given that hybridization is not rare at all in plants

Response: We followed this advice and modified accordingly.

Originally: *While hybrids are usually rare, and may be unfit, back-crossing with parental lineages can enable transfer of adaptively relevant alleles between lineages that otherwise remain distinct^{1,2}.*

Revised (Introduction, lines 33-35): *While hybrids are usually rare in vertebrates, and may be unfit, back-crossing with parental lineages can enable transfer of adaptively relevant alleles between lineages that otherwise remain distinct^{1,2}.*

Line 36 - I think being careful with wording is important here. Hybrid lineage implies hybrid speciation, which is likely rare per Schumer et al. Instead, I think it would make more sense to refer to these lineages as ones within which extensive hybridization has happened. This may sound like semantics, but hybrid speciation implies a certain process and that hybridization itself led to reproductive isolation. This is like rare in general, as you note. The evidence for hybridization having occurred at some point during the evolutionary history of a given lineage, however, is growing and it seems like hybridization has played a role in the evolutionary history of many species.

Response: As outlined in our reply to the reviewer's comment on the title, we fully agree that our data cannot identify if hybridization contributed to reproductive isolation. In fact, we consistently avoid the term 'hybrid speciation' to not give the impression that the hybrid event itself is a cause of reproductive isolation (something that is basically impossible to show with genomic data alone, not to mention that reproductive isolation is not an all-or-nothing phenomenon). In this, we believe that we are in full agreement with the reviewer.

However, we disagree with the reviewer's statement that 'hybrid lineage implies hybrid speciation'. In fact, in our manuscript, we have taken care to ensure that this inference should not be drawn. Firstly, we introduce the term 'hybrid lineage' to refer back to the previous sentence, i.e., that these lineages are "*characterised by mosaic genomes with major contributions from two or more parental taxa*" (lines 27, 28). That is, a hybrid lineage simply is a lineage with an (extensively) admixed genome. It does not imply anything about the process by which the genomes became highly admixed, or why they have been maintained like that. Secondly, as pointed out above, we do not make any inference about the causes of reproductive isolation. It is possible that hybridization contributed one way or another to reproductive isolation, but we

simply cannot know and therefore adopt a terminology that is silent on this issue, and do not speculate further.

In summary, we believe that it is rather uncontroversial to use 'hybrid lineage' to refer to mosaic genomes with major contributions from two or more taxa, and that this usage of the term does not imply anything about the speciation process. We have therefore kept this wording.

Line 43 - 44 - I am not familiar with the flora and fauna of this region, but it would surprise me if no papers exist that show hybridization in the evolutionary history of a lineage.

Response: This statement was not meant to imply that hybridization has not been documented for any Mediterranean lineage. However, we aimed to make a qualitative statement about the relative importance of such events. We have rephrased this sentence to clarify what we mean.

Originally: *Still, it remains to be shown if introgressive hybridization has contributed to the exceptional levels of biodiversity and endemism of the Mediterranean fauna.*

Revised (Introduction, lines 44-46): *Still, it remains to be seen to what extent introgressive hybridization has contributed to the exceptional levels of biodiversity and endemism of the Mediterranean fauna.*

Line 80 - Why use SNV when SNP is so much more widespread in the literature?

Response: Both single nucleotide variant (SNV) and single nucleotide polymorphism (SNP) are used in genomic studies. In our opinion, SNP is most commonly used in the context of allele polymorphisms at the population level, while SNV represents site variations at all levels. SNV is also frequently used in similar phylogenomic studies with a similar scope to our study (e.g., Figueiro et al. 2017; Wang et al. 2020). We therefore prefer to follow suit use the term SNV throughout the manuscript.

Line 200 - 201 - or structural rearrangements, right?

Response: This is a good point, and it is correct that also inversions, for example, can help to maintain unusually long genomic blocks of different ancestry. We therefore qualified our statement accordingly.

Originally: *Unusually long genomic blocks of different ancestry that contain genes are therefore putative candidates for adaptive introgression.*

Revised (Results, lines 218-220): *Unusually long genomic blocks of different ancestry may therefore be putative candidates for adaptive introgression.*

Fig 4 - some font is too small to read

Response: Reviewer #1 raised a similar concern and we have modified accordingly.

Line 236 - panel d, right?

Response: Thank you for spotting this mistake. We have now corrected it.

Reviewers' Comments:

Reviewer #1:

Remarks to the Author:

The authors have implemented all of my comments very well and to my understanding also those of the other reviewers. I really enjoyed reading this paper and I think it will make a great contribution to Nature Communications. Congratulations to the authors. I have no major comments on the analyses or interpretation left. However, I would like to encourage the authors to make the code used publicly available and I have one tiny comment: The «%» sign on PC2 variation explained is missing in the Extended Data Fig. 1.

Reviewer #2:

Remarks to the Author:

The authors have done a thorough job of responding to the few issues raised by reviewers, including myself. I particularly like the addition of the dN/dS analyses of genes within the introgressed block (in response to reviewer 1). In the few cases where authors did not follow suggestions, I thought their reasoning to be sound.

In all, the revisions have improved on an already very strong submission. I look forward to seeing this in print!

Well done.

Reviewer #3:

Remarks to the Author:

I think the authors have done a great job at addressing reviewer comments and have no further suggested changes to the manuscript.

Below is the report of our manuscript revision. We show sentences taken from the manuscript in italic, and parts newly inserted into the manuscript as underlined. Page and line numbers refer to positions in the document with tracked changes.

REVIEWER COMMENTS

Reviewer #1 (Remarks to the Author):

The authors have implemented all of my comments very well and to my understanding also those of the other reviewers. I really enjoyed reading this paper and I think it will make a great contribution to Nature Communications. Congratulations to the authors. I have no major comments on the analyses or interpretation left. However, I would like to encourage the authors to make the code used publicly available and I have one tiny comment: The «%» sign on PC2 variation explained is missing in the Extended Data Fig. 1.

Response: We thank the reviewer for the thorough review of our manuscript and for the positive evaluation. We have corrected the minor error in the PCA plot, thank you for pointing this out.

Reviewer #2 (Remarks to the Author):

The authors have done a thorough job of responding to the few issues raised by reviewers, including myself. I particularly like the addition of the dN/dS analyses of genes within the introgressed block (in response to reviewer 1). In the few cases where authors did not follow suggestions, I thought their reasoning to be sound.

In all, the revisions have improved on an already very strong submission. I look forward to seeing this in print!

Well done.

Response: We are grateful for the strong endorsement and appreciation of our study.

Reviewer #3 (Remarks to the Author):

I think the authors have done a great job at addressing reviewer comments and have no further suggested changes to the manuscript.

Response: We thank the reviewer for the overall positive evaluation and for the constructive feedback.